# Compensation machining of freeform based on fast tool servo system

**Junfeng Liu, Yuqian Zhao● \*, Kexian Liu, Linfeng Wang, Fei Li**

National University of Defense Technology, Changsha, 410000, China

* zhaoyuqianqq11@163.com

## Abstract

With the advantages of large field of view, low cost and simple structure, the freeform optical system has extensive requirements in space exploration and other fields. However, the current machining methods for freeform are difficult to meet the requirements of optical use. Based on a developed fast servo tool (FTS) device, this paper proposes an error compensation turning method for freeforms. Firstly, the Zernike polynomial fitting method is used to reconstruct the freeform surface shape error obtained from off-line measurement, and the offset compensation is used to correct the tool path. Then, the compensation processing physical system is built to simulate the off-line compensation processing of the workpiece to verify the feasibility of compensation processing. Finally, the turning compensation processing of convex freeform aluminum mirror is carried out, and the surface accuracy of the workpiece meets the requirements of visible band. The research results have important practical significance for realizing the fast response machining of free-form surface mirror.

**Data Availability Statement:** All relevant data are within the manuscript and its Supporting Information files.

## Introduction

ADue to its special surface shape and multi degree of freedom design concept, the freeform optical system is able to use fewer optical elements to achieve higher optical efficiency. Compared with traditional optical systems, it has the advantages of low cost, simple structure, high optical performance, etc., and is widely used in civil, military and space exploration fields [1,2]. The supporting structure of the optical system and the mirror body can adopt the same material to form a non heating optical mechanical integrated optical system, so as to avoid the thermal stress and strain on the optical mechanical interface caused by the mismatch of expansion coefficient, and improve the stability of the performance of the optical system [3]. Aluminum materials have attracted much attention in optical systems because of their low cost, light weight and good machinability [4,5]. At present, the aluminum mirror processed by ultra-precision machine tools can meet the use band of near-infrared light [6]. How to further improve its surface accuracy to expand its band range has become the focus and difficulty of research at home and abroad.

With the improvement of the requirements of the optical system on the shape and surface quality of aluminum mirrors, the rapid response manufacturing process combining ultra precision turning and polishing is the mainstream method today [7]. In the process of ultra

**Funding:** This work is financially supported by National Natural Science Foundation of China (No.51991372).

**Competing interests:** The authors have declared that no competing interests exist.

precision turning, the non rotational symmetrical surface error caused by machine tool motion error and workpiece dynamic balance is difficult to be eliminated by two-axis turning. The FTS technology with the capability of non rotational symmetrical surface processing can be applied to the compensation processing of such surface error, so as to realize the direct forming of near-infrared band mirror, which improves the machining accuracy and efficiency of ultra precision turning [8].

In the 1980s, Lawrence National Laboratory of micro feed technology first developed micro piezoelectric knife drive system in the United States. After more than 30 years of development, a series of high-performance micro nano feed systems for FTS and a set of mature manufacturing theory and processing technology system have been developed [9–13]. With the influence of fatigue reliability, Liu J F et al. developed a high-frequency response and large stroke fast knife servo device, and machined a submicron precision microstructure array on the cylindrical surface [14,15].

Based on of a three-axis ultra precision lathe, Zou X C compensated the over cut convex sphere with a curvature radius of 250mm and a diameter of 70mm, and the surface shape accuracy of the workpiece was improved by 39.19% [16]; Using a FTS system with the stroke of 7.5μm and the servo bandwidth of 100 Hz, the Korean robotics and manufacturing technology center measured and compensated the axial error of the machine tool in real time, and processed a large aspheric off-axis aluminum mirror with the diameter of 620mm and the shape accuracy of 0.7μm [17]; Gao W et al. measured and analyzed the contour error in the vector height direction of the surface caused by the error movement of the x-axis and the spindle, and adopted the developed FTS system to compensate the large sinusoidal ray surface, whose PV value was reduced from 0.27 μm to 0.12 μm [18].

The researches above deeply investigated various FTS systems and their applications in compensation processing of aluminum mirrors, and the accuracy of workpiece surface shape meets the requirements of near-infrared band. In order to further improve its surface shape accuracy and reach the application range of visible light, this paper starts with the surface error reconstruction method of free-form surface, and uses the developed FTS [14,15] system to compensate the error corresponding to specific items, so as to improve the machining efficiency and accuracy of ultra precision turning, which also has important guiding significance for improving the machining accuracy of other types of complex surfaces.

## Error analysis of ultra precision turning

The error in ultra precision turning can be divided into in-situ error and off-line error [19]. In situ error refers to the error generated in the processing process, while off-line error refers to the deformation error caused by clamping deformation and processing residual stress after processing. The mapping relationship between the error source and the final shape error is analyzed, and the surface shape distribution characteristics of the shape error caused by the main factors are obtained, which provides a theoretical basis for selecting specific items for error reconstruction when using Zernike polynomial. The frequency components of surface error caused by nonlinear factors such as cutting vibration and environmental instability are complex and irregular, and can be improved by optimizing cutting parameters and environmental conditions. Therefore, the distribution characteristics of surface error caused by centrifugal force and workpiece weight are mainly discussed here.

### Centrifugal force deformation error

The error results from the deformation caused by the centrifugal force generated by the rotation of the workpiece is called the centrifugal force deformation error, including radial error

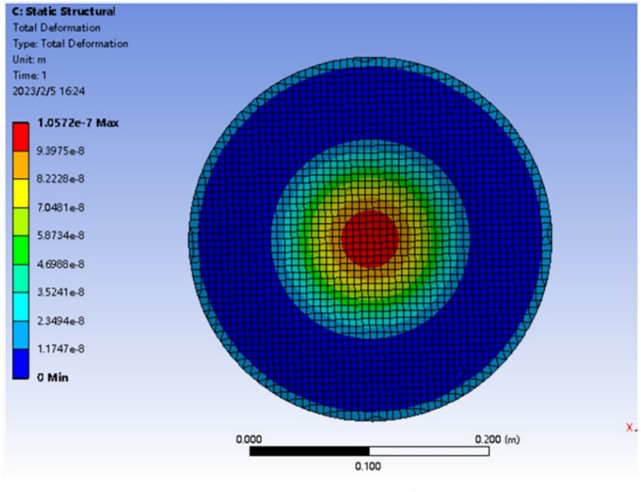 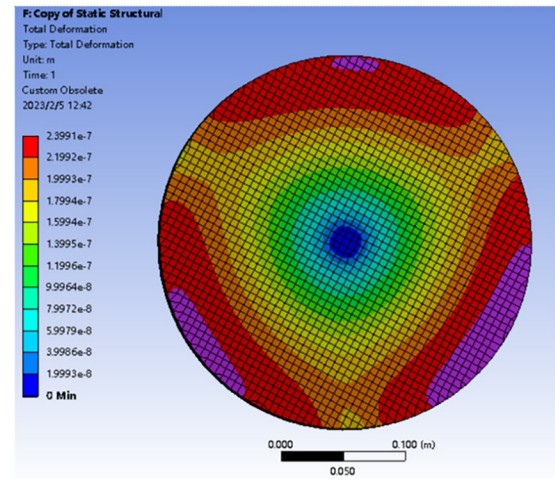

(a) vacuum adsorption (b) outer ring screw compression

**Fig 1. Error distribution of centrifugal force deformation caused by different clamping methods.**

and axial error [19]. Among them, the axial error will be directly reflected in the error along the vector height direction of the workpiece, which has a great impact on the surface shape. The centrifugal stress distribution inside the mirror body is also different when the workpiece is clamped in different ways, so the deformation errors are also different. As shown in Fig 1, for a aluminum alloy spherical mirror with the diameter of 300 mm, the centrifugal force deformation caused by vacuum adsorption clamping method presents the characteristics of lower middle and higher edge, which results in the rotational symmetrical shape error of high center and low edge of the mirror when the centrifugal force disappears. However, the shape error caused by the clamping method of pressing three screws in the outer ring is similar to the form of clover aberration, which is non rotationally symmetrical and difficult to eliminate by using two-axis turning.

## Weight deformation error of workpiece

The deformation error caused by the workpiece weight in horizontal lathe cannot be ignored. The self-weight deformation is shown in Fig 2, and the error distribution is similar to 45° astigmatism.

For freeform surfaces, the deformations caused by centrifugal force and workpiece gravity in the machining process show a non rotational symmetrical distribution, which is difficult to be eliminated by two-axis turning, while the elimination by deterministic modification methods such as magnetorheological polishing seriously affects the machining efficiency.

## Error compensation turning method based on offline detection

In order to improve the accuracy of freeform aluminum mirror surface efficiently, an error compensation turning method based on off-line measurement is proposed. Fig 3 shows the flow of this method. Firstly, the initial surface of the workpiece to be compensated is measured off-line based on the method of fiducial free correction of mapping distortion [20], and the mathematical expression of the initial surface is obtained by Zernike polynomial fitting; the error compensation amount is the difference between the Zernike expressions of the initial surface and the ideal surface. Then the corrected tool path is obtained according to the error compensation. Finally, the micro nano feeding device of FTS is used to realize error

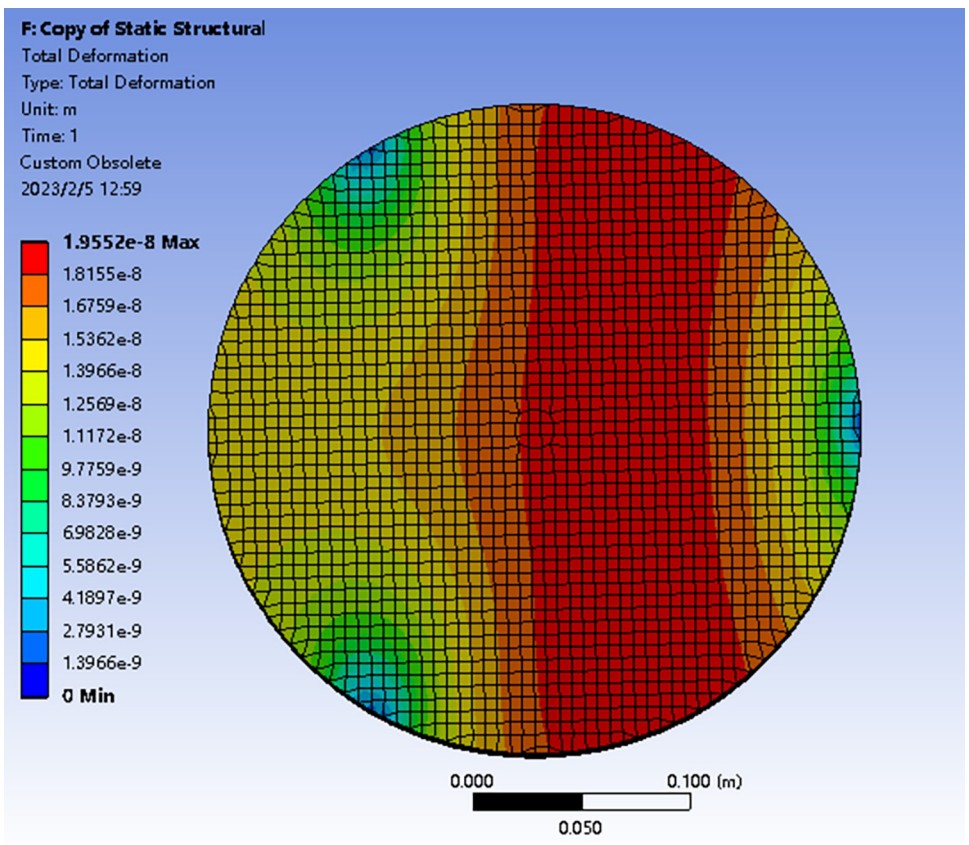

**Fig 2. Simulation results of clamping dead weight deformation of horizontal lathe.**

compensation. Repeat the above steps until the surface accuracy of the workpiece meets the requirements. Inorder to restrain the re-clamping error in offline compensation, a fixture is inserted between the suction cup and the workpiece to reduce the re-clamping deformation, and the positioning accuracy is guaranteed by using the Markov table. It should be noted that the tooling for offline measurement must be consistent with the tooling for processing, so as to reduce the the re-positing error when mounting the workpiece to the machine again after off-line measurement.

## Surface error reconstruction

Zernike polynomials are a set of complete orthogonal bases in the unit circle domain. Using them as the basis function for fitting, the surface shape can be well corresponding to the classical aberration in the optical system. Therefore, the optical imaging performance of the workpiece is able to be intuitively shown by using Zernike polynomial to fit the surface shape error. For a workpiece, its surface shape can be expressed in the form of Zernike polynomial obtained by linear combination of Zernike basis functions:

$$Z(\rho, \theta) = \sum_{n=1}^{M} [\alpha_n Q_n^0(\rho) + \sum_{m=1}^{n} Q_N^M(\rho)\rho^m (b_{nm}\cos(m\theta) + c_{nm}(m\theta))]$$

$$Q_n^m(\rho) = \sum_{s=0}^{n-m} (-1)^s \frac{(2n-m-s)!}{s!(n-s)!(n-m-s)!} \rho^{2(n-m-s)}$$

(1)

Where ($F072,F071$) is the normalized polar coordinate corresponding to the pixel ($u,v$); n and m are polynomial orders; $a_{nm}$, $b_{nm}$ and $c_{nm}$ are polynomial coefficients.

Table 1 lists the first nine Zernike expressions and their corresponding Seidel aberrations.

However, the measured test surface of the workpiece to be compensated is composed of discrete points and cannot be expressed in the form of Formula (1). Assuming the number of discrete points is m, the Zernike fitting expression of the test surface is:

$$\begin{cases} a_1 Z_{11} + a_2 Z_{12} + \ldots + a_n Z_{1n} = W_1 \\ a_1 Z_{m1} + a_2 Z_{m2} + \ldots + a_n Z_{mn} = W_m \end{cases} \tag{2}$$

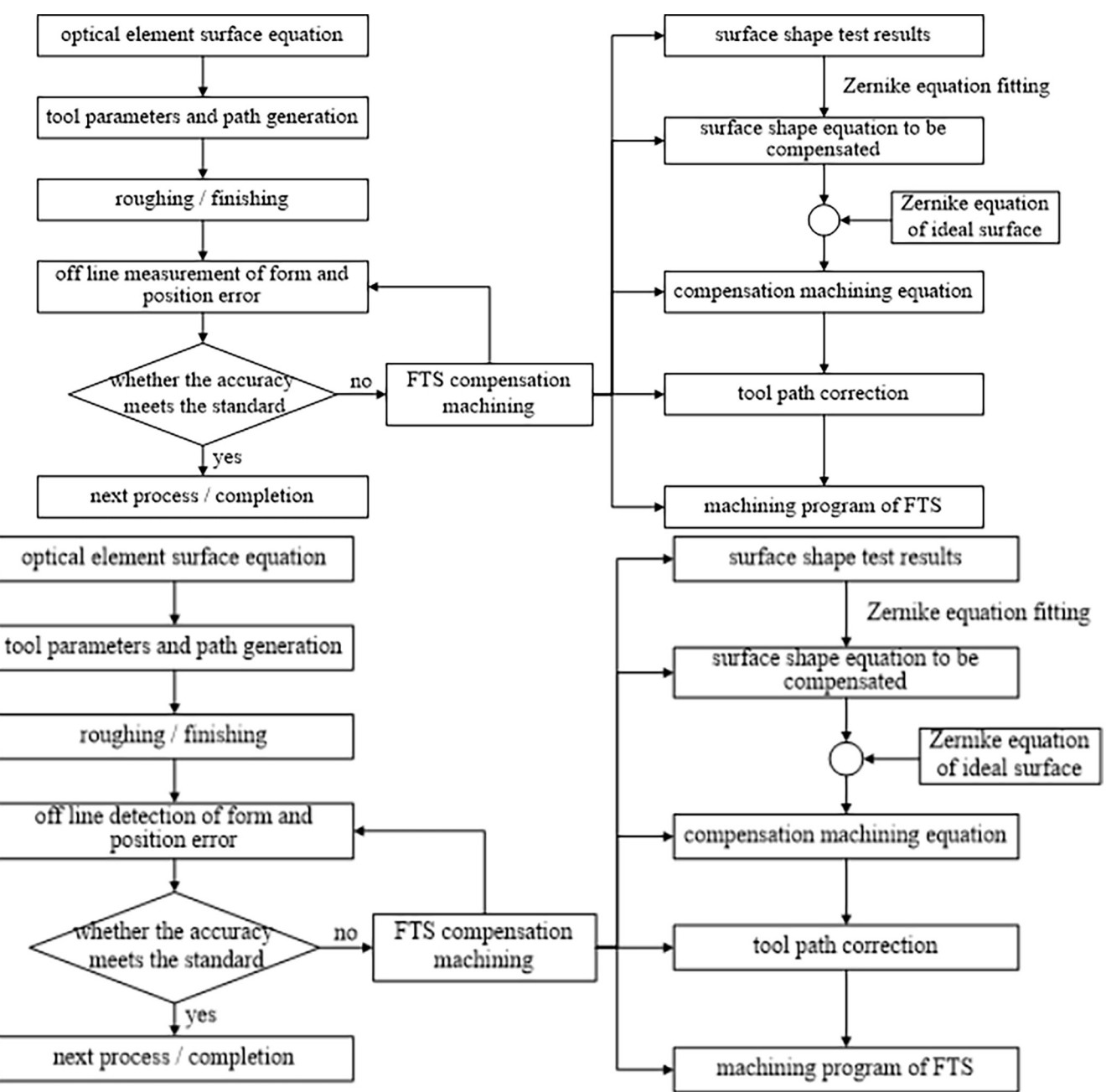

**Fig 3. Flow chart of off-line detection and turning error compensation method.**

**Table 1. Zernike expression and its corresponding Seidel aberration.**

| items | expression | Seidel aberration | items | expression | Seidel aberration |
|---|---|---|---|---|---|
| 0 | 1 | constant term | 5 | $\sqrt{6}\rho^2\cos(2\theta)$ | 0˚ astigmatism |
| 1 | $2\rho\sin(\theta)$ | tilt in Y direction | 6 | $\sqrt{8}(3\rho^2-2)\rho\sin(\theta)$ | coma in Y direction |
| 2 | $2\rho\cos(\theta)$ | tilt in X direction | 7 | $\sqrt{8}(3\rho^2-2)\rho\cos(\theta)$ | coma in X direction |
| 3 | $\sqrt{3}(2\rho^2-1)$ | defocus | 8 | $\sqrt{3}\rho^3\sin(3\theta)$ | clover aberration in Y direction |
| 4 | $\sqrt{6}\rho^2\sin(2\theta)$ | 45˚astigmatism | 9 | $\sqrt{8}\rho^3\cos(3\theta)$ | clover aberration in X direction |

Where $W_i = W(F072_i, F071_i)$ is the vector height of the *i*-th measuring point; $Z_{ij} = Z_i(F072_j, F071_j)$ is the measured value of the *j*-th term of Zernike polynomial at point *i*. The above formula can be changed into:

$$ZA = W \tag{3}$$

Where $Z = (Z_{ij})_{m\times n}$, $A = (a_1, a_2,...,a_n)^T$, $W = (W_1, W_2,...,W_m)^T$.

The purpose of Zernike fitting is to solve the coefficient vector *A*. Using the least square method, *A* can be obtained by Eq (4):

$$Z^T W = Z^T ZA \tag{4}$$

After obtaining the coefficient vector *A*, the Zernike fitting expression $Z(F072,F071)$ of the surface shape of the workpiece to be compensated can be obtained according to Eq (1). If the Zernike expression of the ideal surface shape of the workpiece is $Z_0(F072,F071)$, the compensated machining quantity can be obtained, which is also an expression in the form of Zernike polynomial:

$$z(\rho, \theta) = Z(\rho, \theta) - Z_0(\rho, \theta) \tag{5}$$

Since the compensated machining quantity cannot be negative, it must be ensured that the expression of the compensated machining quantity is greater than or equal to zero within the range of $Z(F072,F071)$, and the obtained $z(F072,F071)$ is offset to obtain the compensated machining quantity expression in line with the actual machining is as follows:

$$z^t(\rho, \theta) = z(\rho, \theta), \min(z(\rho, \theta)) \geq 0$$
$$z^t(\rho, \theta) = z(\rho, \theta) + |\min(z(\rho, \theta))|, \min(z(\rho, \theta)) < 0 \tag{6}$$

## Tool path correction

The tool path can be corrected as shown in Fig 4 according to Eq (7). The actual path is the curve of the last contact point between the tool and the workpiece, while the ideal path is the curve of the ideal contact point between the tool and the workpiece, that is, the projection line of the design surface on the cutting plane. It is easy to get that $P_cP'$ is the compensation amount in the vector height direction, that is, the vector height value of $z(F072,F071)$ at $P'$, and $PP'$ is the compensation amount in the normal direction of the ideal curve. Finally, the relationship between $P_cP'$ and $PP'$ is:

$$|P_cP'| = |PP'|/\cos\alpha \tag{7}$$

When α changes between 0˚ and 30˚, it can be approximately considered that $|P_cP'| = |PP'|$ [21]. Therefore, a new machining path can be obtained by the compensated machining amount $z(F072,F071)$.

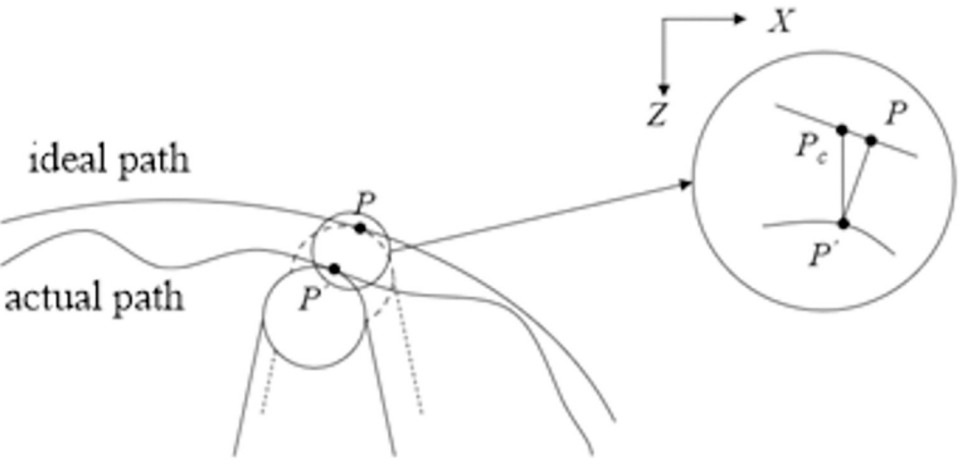

**Fig 4. The diagram of tool path correction.**

### Construction of offline simulation compensation processing system

To verify the feasibility of compensation processing method, the offline simulation compensation machining system is designed and built based on the surface error reconstruction and tool path correction. As shown in Fig 5, firstly, the PMAC controller receives the position information $F072$ and $F071$ obtained from the $X$-axis and $C$-axis of the machine tool and obtains the feed value $z(F072,F071)$ of FTS device [14,15] at this time according to the program downloaded from the upper computer, and then inputs the voltage control signal to the FTS device according to the corresponding relationship between the piezoelectric ceramic displacement $z$ and the input voltage $U$. After being amplifed by the driver, the generated input voltage signal drives the FTS device to output displacement. The Kearns laser sensor is responsible for measuring the actual output displacement of the cutter and inputting it into the data acquisition system to guide the subsequent iterative compensation processing.

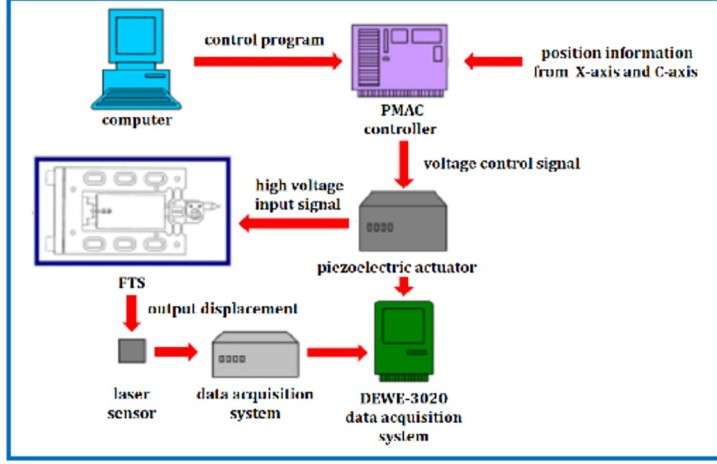

(a) sketch map

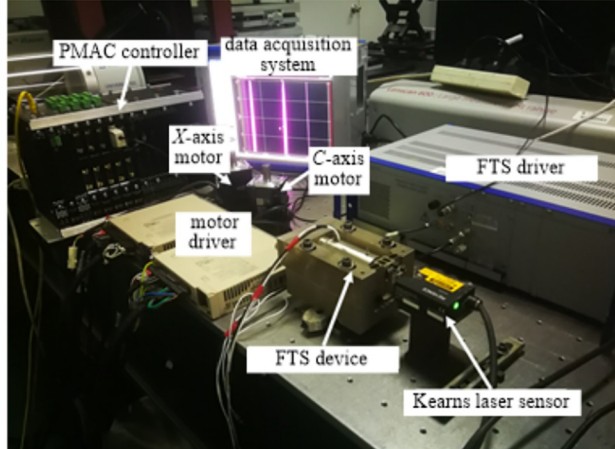

(b) physical map

**Fig 5. Offline simulation compensation machining system.**

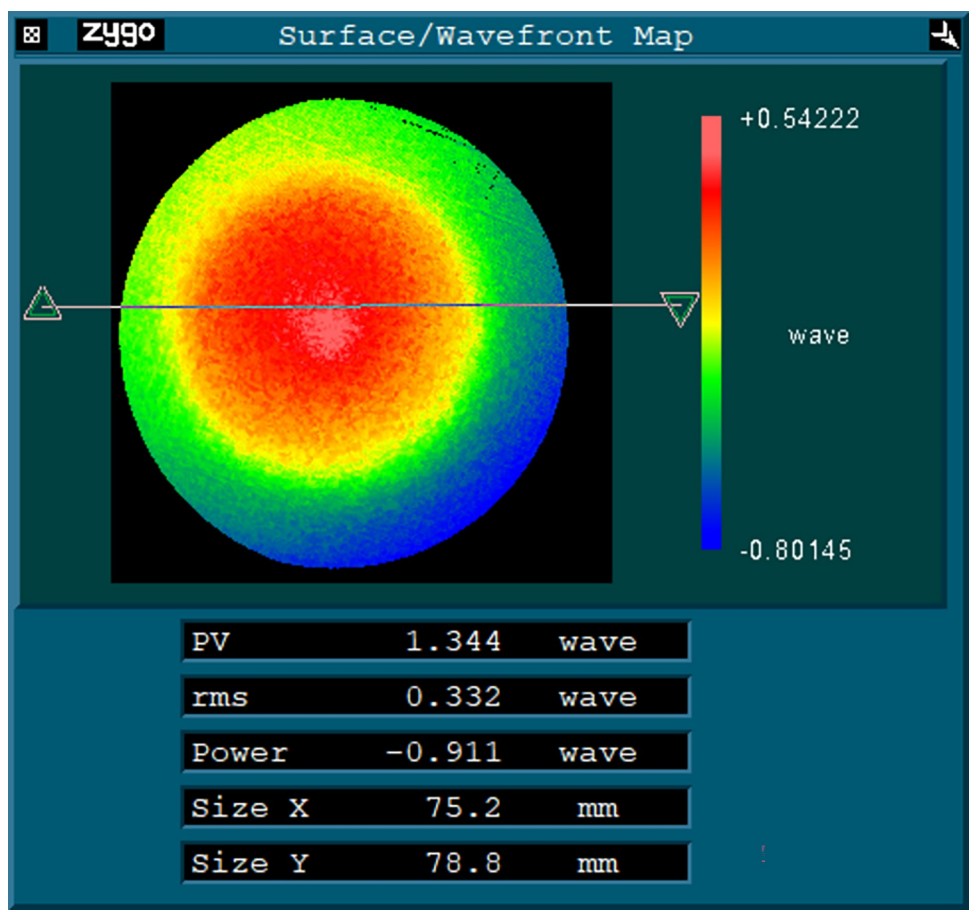

**Fig 6. Initial surface error.**

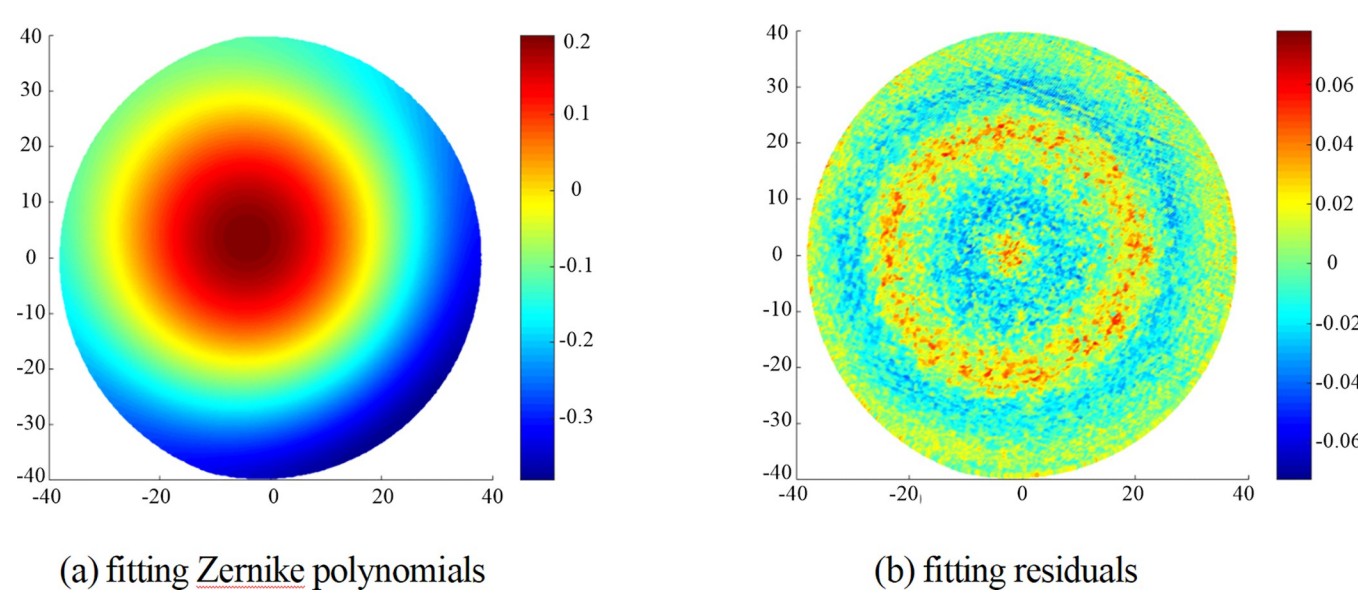

(a) fitting Zernike polynomials          (b) fitting residuals

**Fig 7. The fitting Zernike polynomials and fitting residuals.**

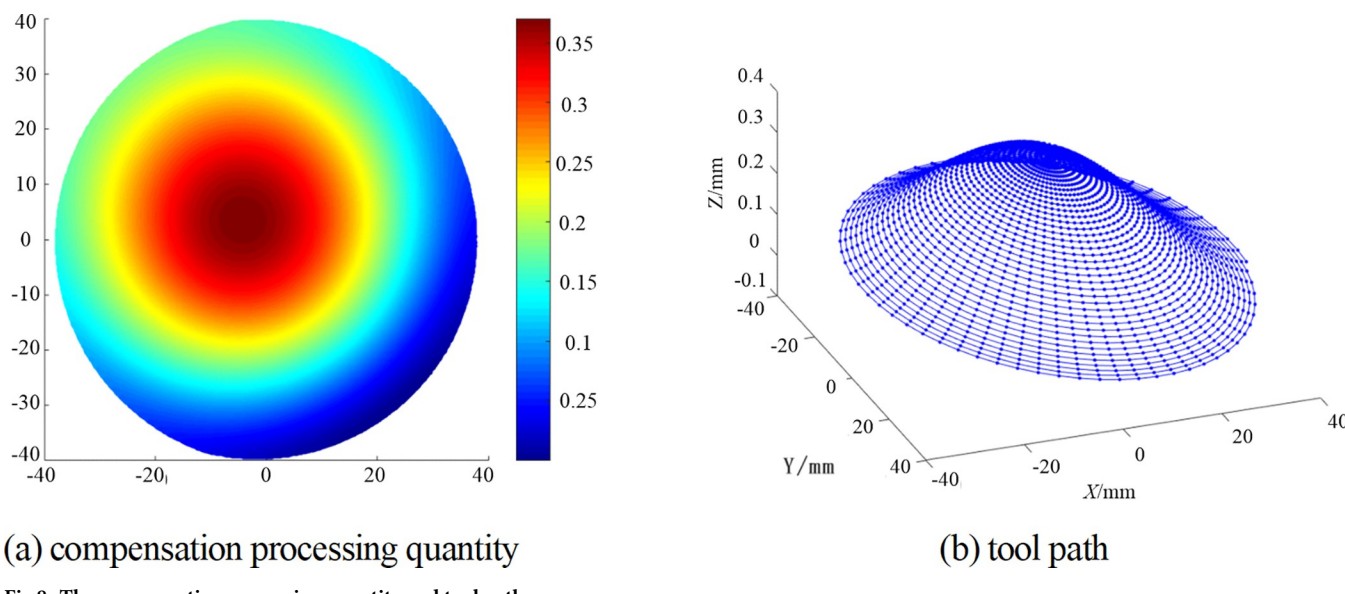

(a) compensation processing quantity

(b) tool path

**Fig 8. The compensation processing quantity and tool path.**

## Simulation compensation machining experiment

Fig 6 shows the result of the surface error of the workpiece to be compensated measured offline by the interferometer (ZYGO VeriFire MST) based on the method of fiducial free correction of mapping distortion [20]. The error distribution is non rotationally symmetrical, which is difficult to be eliminated by two-axis ultra precision turning. The values of PV and RMS of the surface shape are 1.344 λ and 0.332λ respectively, which cannot be directly used in near-infrared and visible optical systems. The error surface is reconstructed based on Zernike polynomial, and the fitting surface equation is as follows:

$$z(\rho, \theta) = 0.002551\sin(\theta)(-4.78e - 5\rho^3 + 0.05033) - 0.03155\cos(\theta)(-4.78e - 5\rho^3 + 0.05033) - 8.501e - 7\rho^2\cos(2\theta) - 0.003564\rho\cos(\theta) + 4.723e - 6\rho^2\sin(2\theta) \tag{8}$$
$$+ 0.003759\rho\sin(\theta) - 0.0006266\rho^2 + 1.775e - 7\rho^4 + 0.7242$$

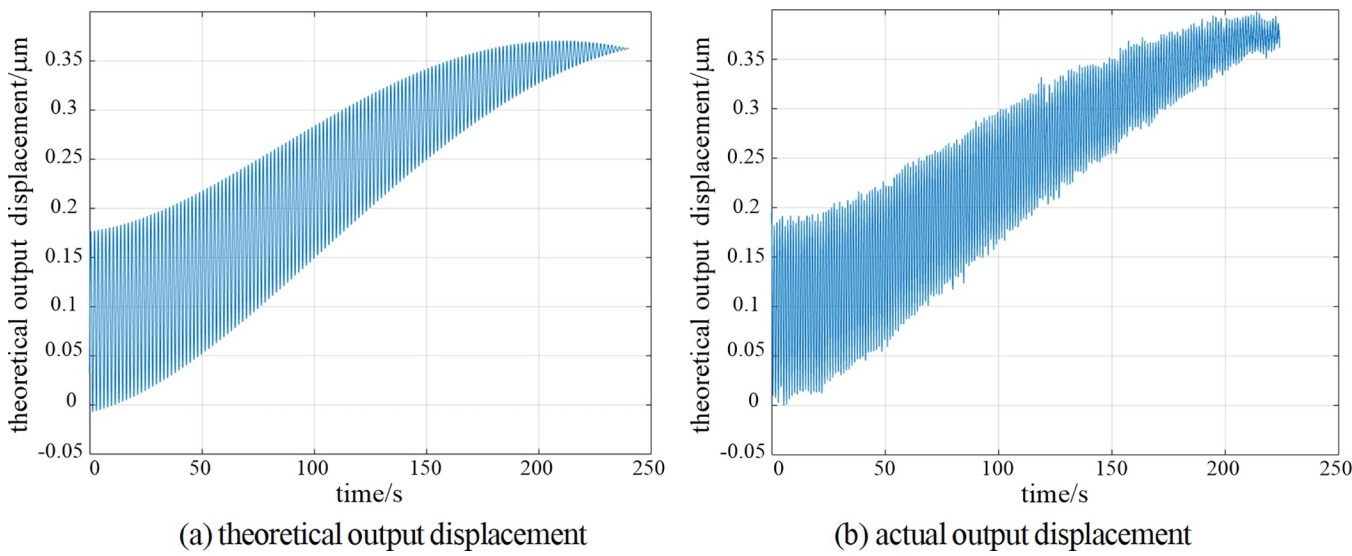

(a) theoretical output displacement

(b) actual output displacement

**Fig 9. The theoretical and actual output displacements.**

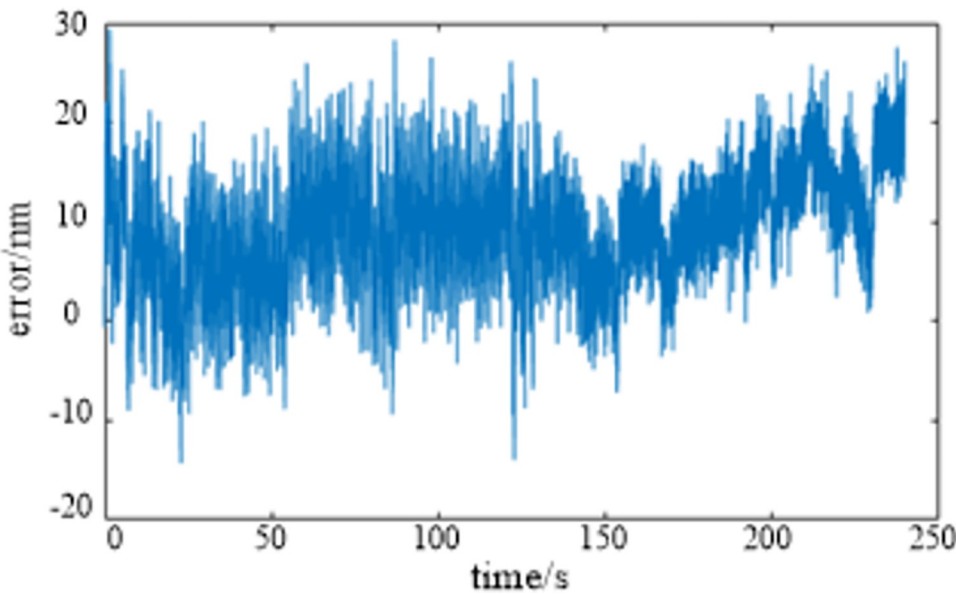

**Fig 10. The curve of path tracking error.**

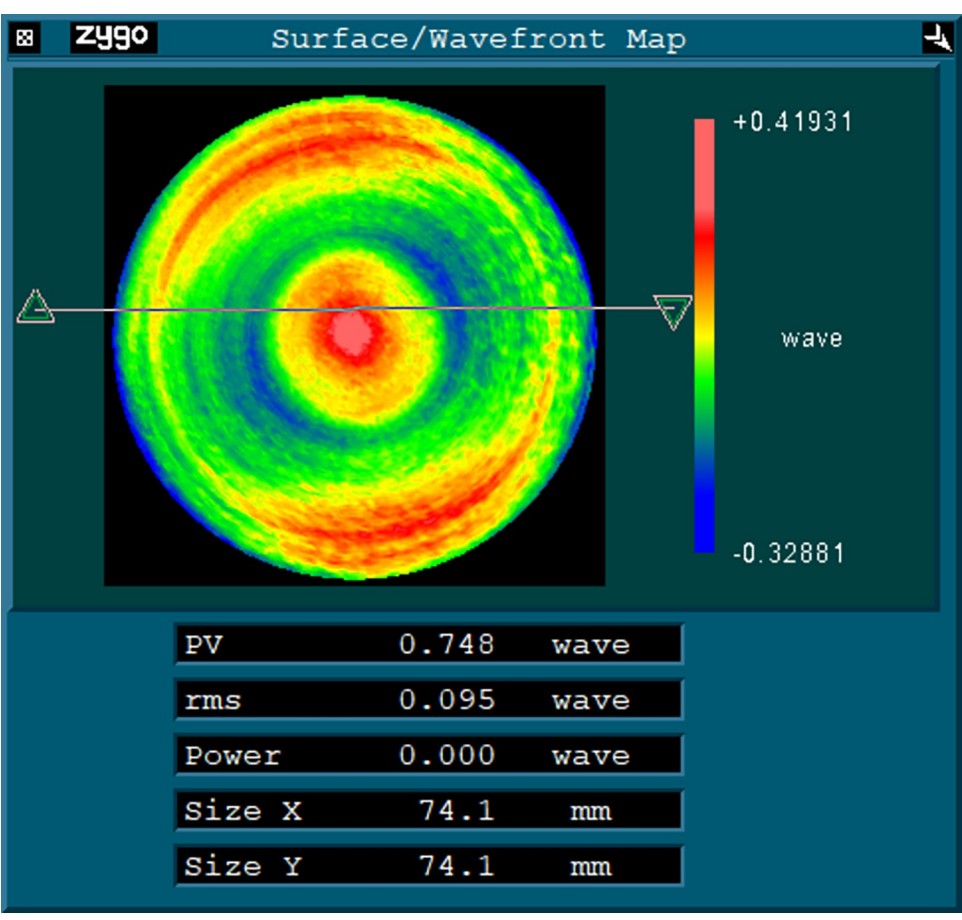

**Fig 11. The surface error.**

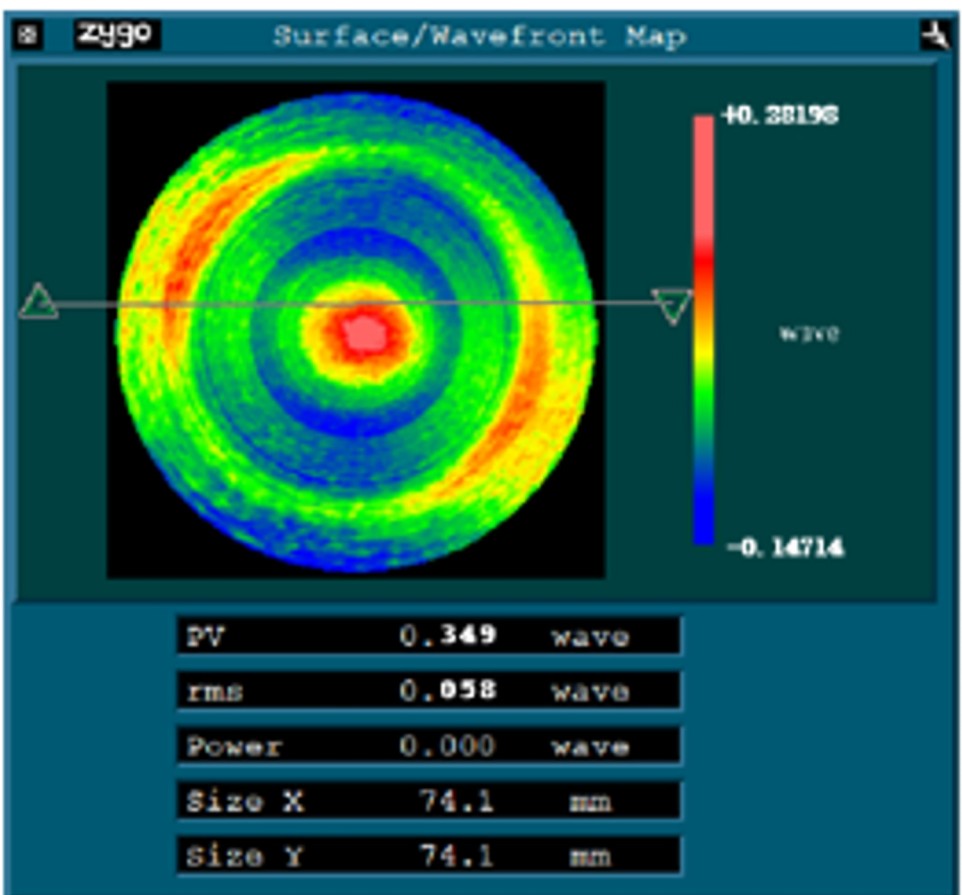

**Fig 12. The result of the first compensation processing.**

The fitting Zernike polynomials and residuals are shown in Fig 7, and the values of PV and RMS are 0.154 λ and 0.039λ respectively. The compensation machining amount is obtained by offsetting the fitting results of Zernike polynomials, and the tool path can be obtained according to Section 3.2, as shown in Fig 8.

The simulation compensation processing of the amount to be compensated is carried out based on the offline simulation compensation machining system. The rotation speed of axis $C$ is 100 rpm, the feed speed of axis $X$ is 10 mm/min, and the radius of the workpiece to be compensated is 40 mm. The variation curves of theoretical and actual output compensations with time are shown in Fig 9.

The theoretical path tracking error curve shown in Fig 10 can be obtained by making a difference between the two curves. The error is between -10 nm to 30 nm, the peak valley value is about 40 nm, and the root mean square value is about 10 nm. The theoretical processing quantity can be basically realized, which proves the rationality of the compensation processing algorithm. However, there are many burrs in the actual output displacement curve, meanwhile the deviation of some output points is large, and the deviation in the tail section is more obvious. This is mainly caused by the noise and accuracy of the two servo motors simulating the $X$-axis and $C$-axis of the machine tool. In the subsequent actual processing, the processing environment and the accuracy of the two axes will be improved, and this phenomenon will also be improved.

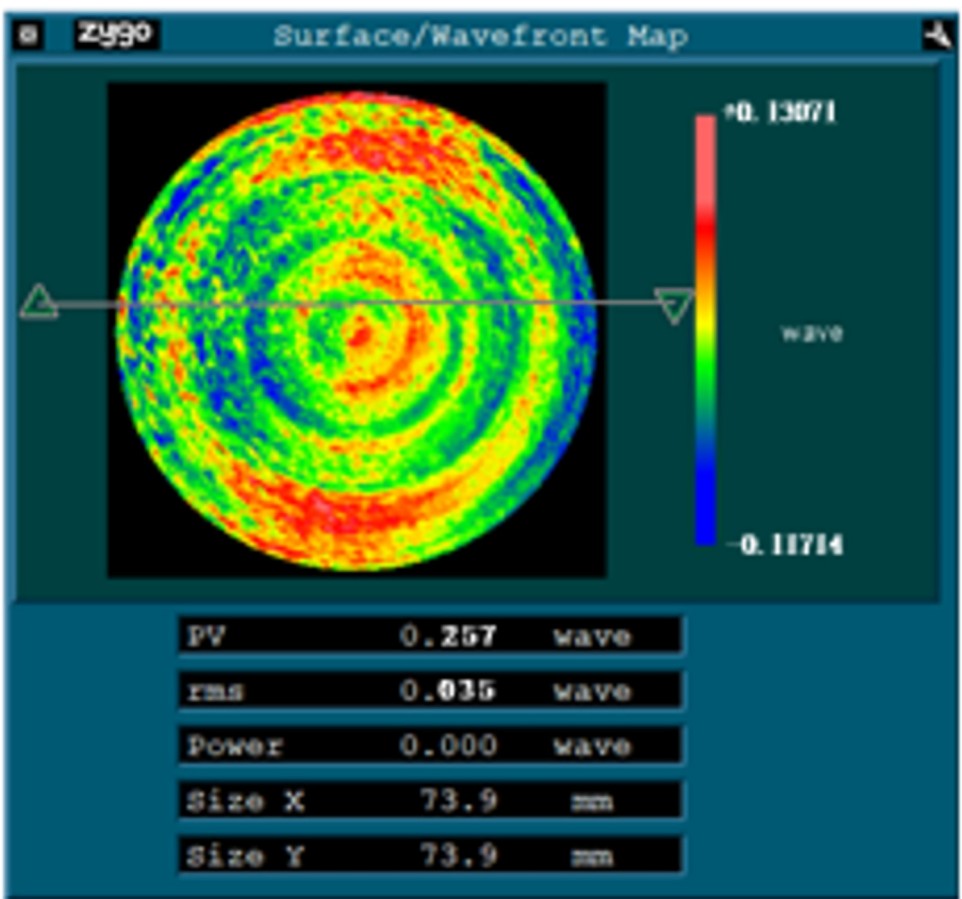

**Fig 13. The result of second compensation processing.**

## Application experiment of compensation machining

A convex spherical workpiece with diameter of 74mm and curvature radius of 250mm is processed for error compensation turning based on the NF350 type ultra precision lathe and identified FTS device. The surface error of workpiece after ultra precision turning is shown in Fig 11. The values of PV and RMS are 0.748 λ and 0.095λ respectively, which is far from meeting the surface shape error requirements of the mirror.

The surface error is reconstructed with Zernike polynomial, and it is found that the astigmatism term coefficient in the 45˚ direction is large, so the fifth term of Zernike fitting coefficient is extracted, and the coefficient is used to construct the compensation processing equation for the first compensation processing experiment. Fig 12 shows the results of the first compensation processing, the astigmatism in the 45˚ direction is reduced, and the values of PV and RMS are reduced to 0.349λ and 0.058λ respectively.

The surface shape after the first compensation processing is reconstructed again, and it is found that the astigmatism term coefficient in the 45˚ direction is still larger. The corresponding coefficient of the fitting result is extracted for compensation processing again, and the processing result is shown in Fig 13. The 45˚ astigmatism is basically eliminated, and the values of PV and RMS are reduced to 0.257λ and 0.035λ respectively.

Repeat the first two steps to reconstruct and compensate the surface shape of 0˚ astigmatism, X-direction coma and X-direction aberration, the surface shape error has been further

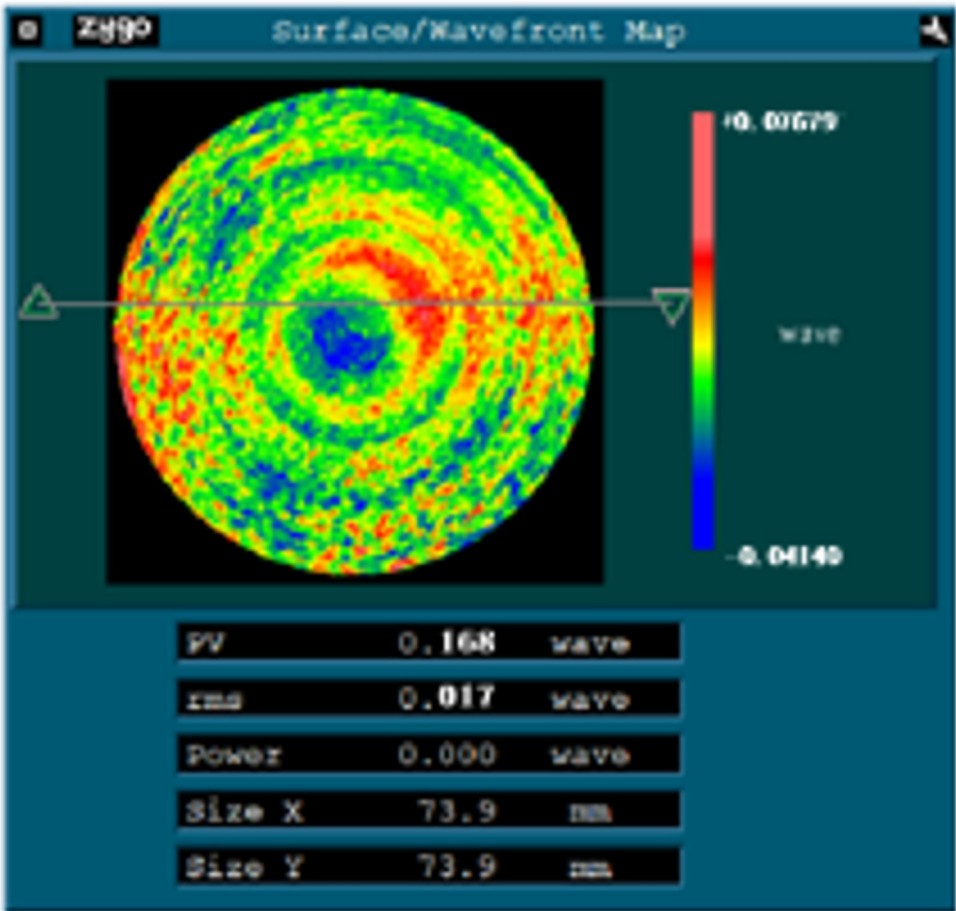

**Fig 14. The result of third compensation processing.**

improved. As shown in Fig 14, the values of PV and RMS are reduced to 0.168λ and 0.017λ respectively.

After three times of compensation processing, the shape accuracy of aluminum mirror nearly meets the requirements of visible light band (PV<1/6λ, RMS<1/50λ), and the turning accuracy is greatly improved. At the same time, compared with other processes, such as fine polishing, the processing efficiency has been effectively improved. Fig 15 presents the processing site and test workpiece.

## Conclusions

With the use of the reconstruction of surface error, a method for error compensation turning is investigated, and the offline and online verification experiments are performed respectively. From the results the following conclusions are drawn:

a. In the process of machining, the workpiece deformation caused by centrifugal force and workpiece gravity is likely to be non rotationally symmetrical, and it is difficult to eliminate it by two-axis turning.

b. Zernike polynomial fitting of surface error can obtain the compensated machining quantity, which can be used as the basis of tool path correction.

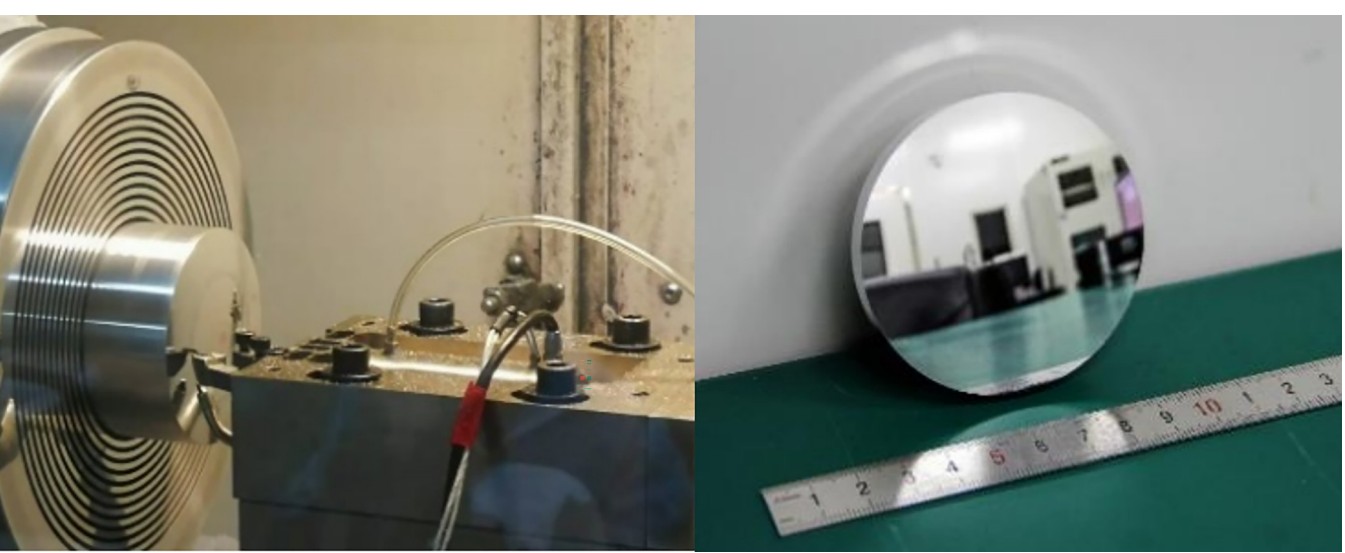

**Fig 15. The processing site and test workpiece.**

c. The off-line compensation processing system is built, and the rationality of the compensation processing method is verified.

d. Through three times of compensation turning, the surface accuracy of freeform aluminum mirror is close to meeting the requirements of visible light, which greatly improves the accuracy and efficiency of aluminum mirror processing.

## Supporting information

**S1 Data.**
(ZIP)

**S1 File.**
(WBPJ)

## Author Contributions

**Investigation:** Junfeng Liu, Yuqian Zhao, Kexian Liu, Linfeng Wang, Fei Li.

**Methodology:** Junfeng Liu.

**Writing – original draft:** Junfeng Liu.

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
