## [Decision Letter · Decision Letter 0]

19 Dec 2022

PONE-D-22-29534Compensation machining of freeform based on fast tool servo systemPLOS ONE

Dear Dr. Zhao,

Thank you for submitting your manuscript to PLOS ONE. After careful consideration, we feel that it has merit but does not fully meet PLOS ONE’s publication criteria as it currently stands. Therefore, we invite you to submit a revised version of the manuscript that addresses the points raised during the review process.

We look forward to receiving your revised manuscript.

Kind regards,

André Gustavo de Sousa Galdino

Academic Editor

PLOS ONE

Journal Requirements:

"This work is financially supported by National Natural Science Foundation of China (No.51991372)"

3. Please upload a new copy of Figures 1 (b) and 2 (1)  as the detail is not clear. Please follow the link for more information: https://blogs.plos.org/plos/2019/06/looking-good-tips-for-creating-your-plos-figures-graphics/"" https://blogs.plos.org/plos/2019/06/looking-good-tips-for-creating-your-plos-figures-graphics/

Reviewers' comments:

Reviewer's Responses to Questions

**Comments to the Author**

1. Is the manuscript technically sound, and do the data support the conclusions?

Reviewer #1: Yes

Reviewer #2: Yes

2. Has the statistical analysis been performed appropriately and rigorously? 

Reviewer #1: Yes

Reviewer #2: Yes

3. Have the authors made all data underlying the findings in their manuscript fully available?

Reviewer #1: Yes

Reviewer #2: Yes

4. Is the manuscript presented in an intelligible fashion and written in standard English?

Reviewer #1: Yes

Reviewer #2: Yes

5. Review Comments to the Author

Reviewer #1: 1. Check typos, such as in Section1 "0.7μm" instead of "0.7m", "0.27μm to 0.12μm" instead of "0.27m to 0.12m".

2. Improve the quality of figures, some are really difficult to read.

3. Please discuss the re-positing error when mounting the workpiece to the machine again after off-line measurement.

Reviewer #2: The author would like to publish interesting article entitled "Compensation machining of freeform based on fast tool servo system". Because it presents the compensation machining for freeform based on fast tool servo system, the journal can put it along with current its current stance into current archive.

6. PLOS authors have the option to publish the peer review history of their article (what does this mean?). If published, this will include your full peer review and any attached files.

Reviewer #1: No

Reviewer #2: **Yes: **Professor Dr. Mehmet Serkan Kırgız

---

## [Author Response · Author response to Decision Letter 0]

16 Feb 2023

1. Check typos, such as in Section1 "0.7μm" instead of "0.7m", "0.27μm to 0.12μm" instead of "0.27m to 0.12m".

Responses：The typos are correct，it is format error causes incorrect display.

2. Improve the quality of figures, some are really difficult to read.

Responses：We improved the quality of Figure 1 and 2 according to the opinion of reviewer.

3. Please discuss the re-positing error when mounting the workpiece to the machine again after off-line measurement.

Responses：This is a great idea. At present, we reduce the deformation error between offline measurement and processing tooling by keeping them consistent. Next, we will build an online measurement system to eliminate this kind of error.

---

## [Editor Report · Decision Letter 1]

22 Feb 2023

Compensation machining of freeform based on fast tool servo system

PONE-D-22-29534R1

Dear Dr. Zhao,

We’re pleased to inform you that your manuscript has been judged scientifically suitable for publication and will be formally accepted for publication once it meets all outstanding technical requirements.

Kind regards,

André Gustavo de Sousa Galdino

Academic Editor

PLOS ONE
---

## [Editor Report · Acceptance letter]

8 Mar 2023

PONE-D-22-29534R1 

Compensation machining of freeform based on fast tool servo system 

Dear Dr. Zhao:

I'm pleased to inform you that your manuscript has been deemed suitable for publication in PLOS ONE. Congratulations! Your manuscript is now with our production department. 

Kind regards, 

on behalf of

Dr. André Gustavo de Sousa Galdino 

Academic Editor

PLOS ONE